# Targeted isolation of *Methanobrevibacter* strains from fecal samples expands the cultivated human archaeome

Stefanie Duller[1], Simone Vrbancic[1], Łukasz Szydłowski[2,3], Alexander Mahnert [1,4], Marcus Blohs[1], Michael Predl[5,6], Christina Kumpitsch [1,4], Verena Zrim[7], Christoph Högenauer [8], Tomasz Kosciolek [2,3,9], Ruth A. Schmitz [10], Anna Eberhard[1], Melanie Dragovan[1], Laura Schmidberger[1], Tamara Zurabischvili[1], Viktoria Weinberger[1], Adrian Mathias Moser[8], Dagmar Kolb[11,12], Dominique Pernitsch[11], Rokhsareh Mohammadzadeh[1], Torben Kühnast[1], Thomas Rattei [5] & Christine Moissl-Eichinger [1,4] ✉

Archaea are vital components of the human microbiome, yet their study within the gastrointestinal tract (GIT) is limited by the scarcity of cultured representatives. Our study presents a method for the targeted enrichment and isolation of methanogenic archaea from human fecal samples. The procedure combines methane breath testing, in silico metabolic modeling, media optimization, FACS, dilution series, and genomic sequencing through Nanopore technology. Additional analyzes include the co-cultured bacteriome, comparative genomics of archaeal genomes, functional comparisons, and structure-based protein function prediction of unknown differential traits. Successful establishment of stable archaeal cultures from 14 out of 16 fecal samples yielded nine previously uncultivated strains, eight of which are absent from a recent archaeome genome catalog. Comparative genomic and functional assessments of *Methanobrevibacter smithii* and *Candidatus* Methanobrevibacter intestini strains from individual donors revealed features potentially associated with gastrointestinal diseases. Our work broadens available archaeal representatives for GIT studies, and offers insights into *Candidatus* Methanobrevibacter intestini genomes' adaptability in critical microbiome contexts.

The human gut is a complex ecosystem harboring a diverse community of microbes, including bacteria, fungi, viruses, and archaea[1]. While bacteria are well described, and their role in gut health is well studied, the function and role of archaea in the human gastrointestinal tract (GIT) is still not well understood. Archaea found in the GIT are mainly represented by the order Methanobacteriales, with the most abundant species *Methanobrevibacter smithii* and *Candidatus* Methanobrevibacter intestini[2–4].

[1]D&R Institute of Hygiene, Microbiology and Environmental Medicine, Medical University of Graz, Graz, Austria. [2]Malopolska Centre of Biotechnology, Jagiellonian University in Krakow, Krakow, Poland. [3]Sano Centre for Computational Medicine, Krakow, Poland. [4]BioTechMed Graz, Graz, Austria. [5]Centre for Microbiology and Environmental Systems Science, University of Vienna, Vienna, Austria. [6]Doctoral School Microbiology and Environmental Science, University of Vienna, Vienna, Austria. [7]Center for Medical Research, Medical University of Graz, Graz, Austria. [8]Division of Gastroenterology and Hepatology, Department of Internal Medicine, Medical University of Graz, Graz, Austria. [9]Department of Data Science and Engineering, Silesian University of Technology, Gliwice, Poland. [10]Institute for General Microbiology, Christian Albrechts University, Kiel, Germany. [11]Core Facility Ultrastructure Analysis, Medical University of Graz, Graz, Austria. [12]Gottfried Schatz Research Center, Medical University of Graz, Graz, Austria. ✉e-mail: christine.moissl-eichinger@medunigraz.at

These methanogenic archaea produce methane ($CH_4$) as a metabolic byproduct while producing ATP via methanogenesis[5–9]. The yielded $CH_4$ serves as an important indicator for the presence of methanogenic archaea[10].

$CH_4$ can be detected in humans' breath, as it is partially absorbed into the blood and excreted through the lungs[11]. The impact of $CH_4$ on the human GIT has already been addressed and it has been reported that $CH_4$ leads to the reduced efficiency of intestinal smooth muscle activity, often associated with e.g., the constipation-predominant irritable bowel syndrome[12–14]. Further, variations of $CH_4$ levels in the human breath were discussed as an indicator for enhanced oxidative stress and could therefore show a great potential in the context of in vivo diagnostics[15–17]. On the other hand, positive effects were also described, as $CH_4$ could also act as antioxidant, protecting cells from reactive oxygen species (ROS) and demonstrating potential anti-inflammatory properties[18].

*Methanobrevibacter* species show a prevalence of more than 90% in fecal samples with an average abundance of 0.56% for *Methanobrevibacter smithii* and 0.13% for *Cand.* Methanobrevibacter intestini[3,19]. In high-$CH_4$ producing individuals (up to 40% of the middle European and Northern American population)[20], this average abundance is raised to 2% for *Methanobrevibacter* genus in general[21].

Notably, *Methanobrevibacter* species are highly correlated with certain bacterial representatives of the gut microbiome, e.g., Christensenellaceae, and have been shown to be syntrophically interactive in co-culture experiments with e.g., *Christensenella minuta* or *Bacteroides thetaiotaomicron*[22,23].

Even though (methanogenic) archaea have been shown to be part of the human gut microbiome for more than 40 years[24], knowledge about their function and role within the human gut microbiome is still sparse. The reason for this lies in several methodological issues as standard protocols are mainly bacteria-optimized, which are not optimally working for archaea due to their distinct cell structure, physiology, and metabolic activity. These distinct features and inappropriate protocols make it difficult to visualize, culture or analyze them with multi-omics approaches[25].

Furthermore, the lack of comprehensive reference databases poses a challenge in adequately evaluating methanogenic archaea in the datasets. Additionally, the presence of a strong bacterial or host background further complicates the matter[25]. Therefore, it is of utmost importance to develop new protocols or modify existing ones to effectively study archaea in the complex microbiome environment.

Even though molecular biological techniques help to study the archaeome, isolating and culturing archaea is crucial for in-depth, mechanistic analysis of these microbes.

Based on the Global Catalog of Microorganisms (https://gcm.wdcm.org/) only few human host-associated archaeal isolates are available in culture (three *Methanobrevibacter oralis* strains from human oral cavity and subgingival plaque; and four *Methanobrevibacter smithii* isolated from human feces and large intestine), hindering detailed studies.

To tackle the lack of archaeal cultures for functional studies, we established herein a cultivation procedure to efficiently enrich and isolate methanogenic archaea from human stool samples in a targeted manner. By providing optimal growth conditions (e.g., by providing a suitable bacterial partner), followed by archaea-targeting Fluorescence-Activated Cell Sorting (FACS), dilution series and quality control through Nanopore sequencing, we were able to establish stable methanoarchaeal cultures from 14 out of 16 fecal samples obtained from both healthy and diseased subjects.

By this procedure, we obtained 9 novel methanoarchaeal strains, and were able to retrieve personalized representatives of *Methanobrevibacter smithii* and *Cand.* Methanobrevibacter intestini strains from stool samples of various participants with different health conditions, which were subsequently compared on genomic and functional levels.

As the role of archaea in health and disease is still unknown and it is still discussed whether they might have a potential beneficial or a pathogenic effect on human health, obtaining additional isolates from human samples will allow in-depth analysis, which might help to deepen our understanding of the role of methanogens in the human GIT.

## Results

### Set-up of a cultivation workflow and selection of bacterial syntrophic partners based on in silico predictions

For the cultivation of archaea from human fecal samples, we followed a combined strategy of: i) Providing in silico-customized media, partially including selected bacterial syntrophic partners to facilitate *Methanobrevibacter* growth, ii) pre-selection of sample donors for $CH_4$ exhalation and such a higher methanogen load, iii) rigorous and standardized screening of enrichments for $CH_4$ production and presence of (unique) methanoarchaeal key genes, and iv) FACS and antibiotic treatment for the mechanical and chemical enrichment of methanogens from the bacterial background of syntrophic partners or fecal samples. An overview of this strategy is provided in Fig. 1.

*C. minuta* and *B. thetaiotaomicron* were selected as potential bacterial syntrophic partners, based on previous reports of successfully established co-cultures with *Methanobrevibacter* species[22,23], and the observed co-occurrence in metagenomic datasets[21]. Both species, *C. minuta* and *B. thetaiotaomicron*, were reported to provide hydrogen, acetate and possibly other metabolites to the archaeal partners, with *C. minuta* reported to be more efficient than *B. thetaiotaomicron*[23]. Nevertheless, we performed a more detailed prediction of the metabolic capacities of both bacteria to serve as a potential syntrophic partner for *Methanobrevibacter* species.

Metabolic models were generated for *M. smithii*, *C. minuta* and *B. thetaiotamicron*, based on their genomes. At this early stage, *Cand.* M. intestini was not included in the simulations. Single and co-cultures with MS and MpT1 media were simulated using flux balance analysis (FBA). Across all simulated co-cultures, the growth rate of *M. smithii* was increased compared to the single cultures on the same medium.

This growth boost is enabled by cross-feeding (Supplementary Fig. S5). Multiple metabolites could be identified as mediators of these interactions: The bacteria commonly provided $H_2$, amino acids (alanine, lysine, glutamine, cysteine), nucleobases and nucleosides (uracil, inosine, adenosine, deoxyguanosine, xanthosine), vitamins and coenzymes (menaquinone 7, coenzyme A, NAD), as well as fructose, glycerol, formate, putrescine and ethanol (all output files are provided in our GitHub repository[26]). Based on the simulations, both *B. thetaiotamicron* and *C. minuta* were predicted to be fitting partners for *M. smithii*, without clear preference for either. As such, both bacteria and both media were selected for the enrichment cultures.

To optimize the enrichment cultures, we leveraged the methanoarchaeal autofluorescence of coenzyme $F_{420}$ for FACS[27], a method refined through the use of both stool samples and methanogenic reference strains (see "Materials and Methods"). This approach allowed us to establish effective sorting conditions using FACS, ensuring the accurate enrichment of target methanogens[28].

### Efficient enrichment of novel *Methanobrevibacter* from donors with detectable archaeal microbiome signatures

Fecal samples from all 16 study participants (Supplementary Data S1) were processed for enrichment of methanogenic archaea using suitable media with and without additionally added bacteria (see workflow overview Fig. 1). Overall, 8/8 samples of the healthy cohort and 6/8 of the diseased cohort yielded $CH_4$-positive cultures (88% of all

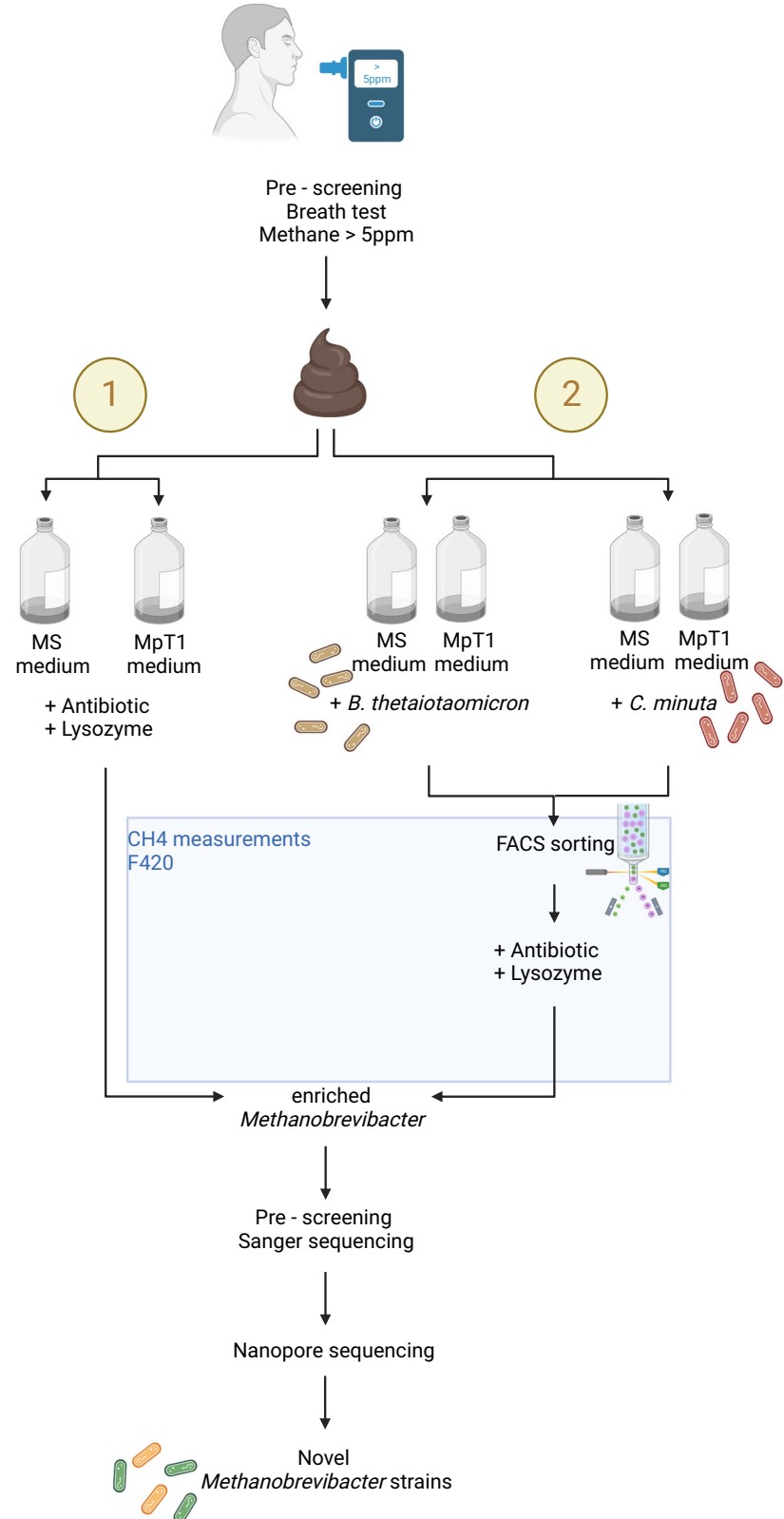

**Fig. 1 | General workflow of the cultivation strategy.** Enrichment process encompassed strategies without (1) and with (2) bacterial partners. Created with BioRender.com.

samples). Some samples yielded several enrichment cultures by different approaches (e.g., P15 and P86).

After initial pre-selection for unique methanoarchaea (according to their *mcrA* gene sequence) of all CH4-positive cultures, enrichment cultures of 14 participant samples were processed further (Table 1). FACS and visual monitoring for potential purity was performed, and after a minimum of 20 transfers they underwent Nanopore-based whole genome sequencing. An overview of the cultivation workflow

**Table 1 | Overview of participants, their health status, the derived cultures and the outcome of the metagenomic profiling of the original fecal sample[a]**

| PARTICIPANTS | | | CULTIVATION | | | METAGENOMIC PROFILING | | | |
|---|---|---|---|---|---|---|---|---|---|
| # | Disorder | breath CH$_4$ [ppm] | Enrichment cultures (medium/syntroph) | Predominant archaeon | CH$_4$ [%] | Bacteria [%] | Archaea [%] | *M. smithii* [% of archaea] | *Cand.* M. intestini [% of archaea] |
| P15 | None | 30 | P15 (MpT1 / B. t.) | *M. smithii* | 16.5 | 99.76 | 0.24 | 88.8 | 5.1 |
|  |  |  | P15 (MS / B. t.) | *M. smithii* | 20.7 |  |  |  |  |
| P20 | None | 17 | P20 (MS / B. t.) | *M. smithii* | 22.2 | 99.82 | 0.18 | 76.5 | 3.5 |
| P31 | None | 70 | P31 (MS / B. t.) | *M. smithii* | 23.6 | n.a. |  |  |  |
| P32 | None | 44 | P32 (MS / B. t.) | *M. smithii* + DS | 23.4 | 99.87 | 0.13 | 90.8 | 6.3 |
| P45 | None | 16 | P45 (MS / B. t.) | *Cand.* M. intestini | 23.8 | 99.84 | 0.16 | 7.9 | 79.5 |
| P46 | None | 21 | P46 (MpT1 / B. t.) | *Cand.* M. intestini | 24.2 | 99.49 | 0.51 | 8.6 | 89.9 |
| P51 | None | 28 | P51 (MS / B. t.) | *Cand.* M. intestini | 5.5 | 99.96 | 0.04 | 6.5 | 83.1 |
| P82 | None | 22 | P82 (MpT1) | *Cand.* M. intestini | 25.2 | n.a. |  |  |  |
| P34 | CD, flatulence, abdominal pain | 22 | n.a. | n.a. | n.a. | n.a. |  |  |  |
| P48 | IBS | 26 | P48 (MpT1 / B. t.) | *M. smithii* | 22.2 | 99.66 | 0.34 | 94.0 | 5.3 |
| P61 | IBS, Lactose I, SIBO | 21 | P61 (MpT1) | *M. smithii* | 23 | n.a. |  |  |  |
| P66 | Lactose I | 55 | P66 (MS / B. t.) | *M. smithii Cand.* M. intestini | 21.8 | 99.16 | 0.84 | 94.6 | 4.8 |
| P85 | CD | 6 | n.a. | n.a. | n.a. | 99.99 | 0.01 | 0 | 0 |
| P86 | Lactose I | 13 | P86 (MS / B. t.) | *Cand.* M. intestini + DS | 22.4 | 99.99 | 0.01 | 63.9 | 3.8 |
|  |  |  | P86 (MS) | *M. smithii* + DS | 21.1 |  |  |  |  |
| P88 | IBS, lactose I, fructose I | 8 | P88 (MS) | *M. smithii* | 22.3 | 99.87 | 0.13 | 72.4 | 3.8 |
| P89 | IBS, lactose I, pancreatic inflam. | 10 | P89 (MS / B. t.) | *Cand.* M. intestini | 21.2 | 99.78 | 0.22 | 5.6 | 88.6 |

[a]CD: Crohn's Disease, IBS: Inflammatory Bowel Syndrome, Lactose I: Lactose intolerance, Fructose I: Fructose intolerance, SIBO: Small Intestinal Bacterial Overgrowth. CH$_4$ analysis indicated the presence of methanogenic archaea in the participant's microbiome. Derived cultures are labeled according to the participant number, the culture medium (MS: MS medium, MpT1: MpT1 medium), and the supplementation with *B. thetaiotaomicron* (B.t.). Additional genomes acquired through dilution series are annotated with + DS. The predominant archaeon as detected through Nanopore sequencing is given, and the outcome of the CH$_4$ measurement of the cultures 10 days of incubation at 37 °C. The metagenomic information of the original fecal sample was analyzed with respect to the relative abundance of Bacteria, Archaea, *M. smithii* and *Cand.* M. intestini.

and sequencing results (including taxonomic classification through kraken2/bracken) for each individual sample is provided on our GitHub repository[26].

At this stage, 12 of 16 enrichment cultures were dominated by *Methanobrevibacter* representatives. In more detail, eight were predominated by *M. smithii*, and three by *Cand.* M. intestini. Further, two bacteria-dominated enrichment cultures (P51 and P46) contained a substantial amount of *Cand.* M. intestini (Fig. 2).

Notably, the dominance of either methanoarchaeon was evident in the original microbiome profiling. Samples were *Cand.* M. intestini was predominant in the profiling, exhibited similar dominance of this microbe in subsequent cultivation attempts. Remarkably, both *Methanobrevibacter* species were present in all samples initially. Yet during the enrichment process, one of these archaea emerged as the predominant species, consistent with their original relative abundance (Table 1). However, in the case of sample P86, we achieved successful enrichment of both *Methanobrevibacter* species separately (Fig. 2).

Noteworthy is the observation that samples P34 and P85 failed to yield archaeal cultures, despite positive methane breath test results of the donors (Table 1). While microbiome profiles were accessible for P85 (P34 had to be excluded due to technical reasons), it was evident that this individual lacked archaeal signatures, suggesting an alternative source of methane production, such as undetected methanogens in the oral cavity or small intestine. It shall be noted that these two individuals were the only ones with Crohn's disease included in this study. It is indeed known that Crohn's disease patients have a reduced amount of archaea in their GIT[29].

To further purify the obtained enrichments and obtain monocultures of the methanogens, subsequent dilution series were and are being performed in combination with consequent antibiotics and lysozyme treatment (see "Materials and Methods"), leading to a substantial improvement of culture purity later on (e.g., P32: from 32.23% *M. smithii* to 95.05%; P 86 (MS + B.t.): from 80.48% *Cand.* M. intestini to 98.4%; Supplementary Data S3). In the meantime, we have improved the purity of more enrichments even further, for more details please visit our GitHub repository[26].

We further visually examined the enrichments with Scanning Electron (SEM) and fluorescence microscopy. The SEM micrographs revealed the presence of mainly short rods (single or in chains) in each of the enrichment cultures, representing the known morphology of *Methanobrevibacter* species. Some rods appeared very bloated, as shown in Fig. 3 (arrows). All cultures were dense and revealed the presence of numerous, actively dividing cells. In agreement with the SEM examinations, we observed cells with *Methanobrevibacter*-typical morphology displaying the F$_{420}$ autofluorescence in all cultures (Supplementary Fig. S6).

**_Bacteroides thetaiotaomicron_, but not _Christensenella minuta_, is a potent syntrophic partner for selective archaeal enrichment**
Methane, being uniquely produced by methanogens, serves as an indicator for their presence in cultures. We noted the presence of CH$_4$ in cultures grown in MS medium with and without supplementation with *B. thetaiotaomicron* after four days of incubation. Similarly, CH$_4$ was detected in cultures grown in MpT1 medium, with or without

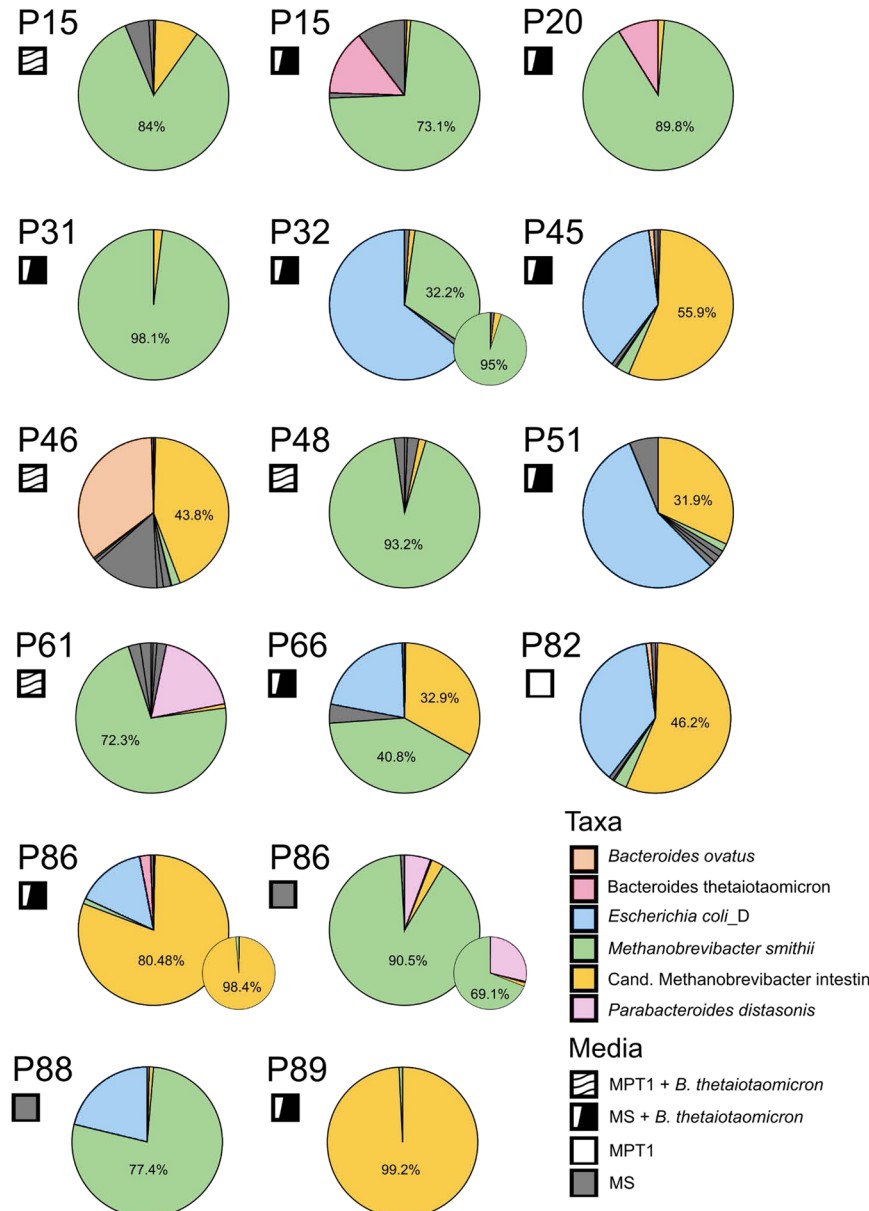

**Fig. 2 | Composition of enrichment cultures.** Classified taxa and their relative abundance as assessed through Nanopore sequencing, followed by Kraken2/bracken with confidence score of 0.3. Small circles show composition after antibiotic treatment and dilutions series for selected cultures, which were ready at the time point of manuscript preparation. *M. smithii* is shown in green and *Cand.* M. intestini in yellow. Percentages of dominating methanogens are provided. All original bracken reports and culture purity updates are provided in our GitHub repository[26]. Source data are provided as a Source Data file.

supplemental *B. thetaiotaomicron* albeit only after 10 days of incubation.

The addition of *B. thetaiotaomicron* influenced our enrichment approach in a distinct manner. While its presence did not affect the overall success rate of methanogen enrichment - evidenced by 12 out of 16 positive enrichment experiments occurring both with and without *B. thetaiotaomicron* - it enabled the isolation of an additional strain (P66) that was not obtained in the abiotic medium (Fig. 2).

In contrast, cultures with additionally added *C. minuta* always remained CH4 negative (up to 8 weeks of incubation). Also, no autofluorescence of coenzyme $F_{420}$ could be detected for these cultivation attempts. This finding was unexpected, as *C. minuta* had been described as an efficient syntrophic partner of *M. smithii*[23] and we had originally expected positive effects in supporting the growth of these methanogens by providing *C. minuta* in the medium.

## Co-enriched bacteria give indications on bacterial partners within the gastrointestinal microbiome

*Methanobrevibacter* species were enriched with background bacteria from the fecal samples. Interestingly, enrichment cultures with a predominance of *M. smithii* (Supplementary Fig. S7) showed only a few significant, weakly positive correlations with specific background bacteria, such as *Phocaeicola dorei* (q = 0.022052; $r_s$ = 0.3641) and UMGS75 sp900538885, a so-far uncultivated Bacilli (q = 1.389*10$^{-9}$; $r_s$ = 0.4242; Spearman correlation and Benjamini Hochberg correction).

A number of negative associations of *M. smithii* were observed, including all *Bacteroides* (e.g., *B. cellulosilyticus*, q = 3.2267*10$^{-6}$; rs = −0.4112) and *E. coli* species (e.g., E. coli_D, q = 7.8638*10$^{-8}$; rs = −0.4926). These were, however, significantly positive with *Cand.* M. intestini (*B. cellulosilyticus*, q = 0.002058; rs = 0.4828; E. coli_D, q = 0.002042; rs = 0.2172; Supplementary Data S4; Supplementary Fig. S7)

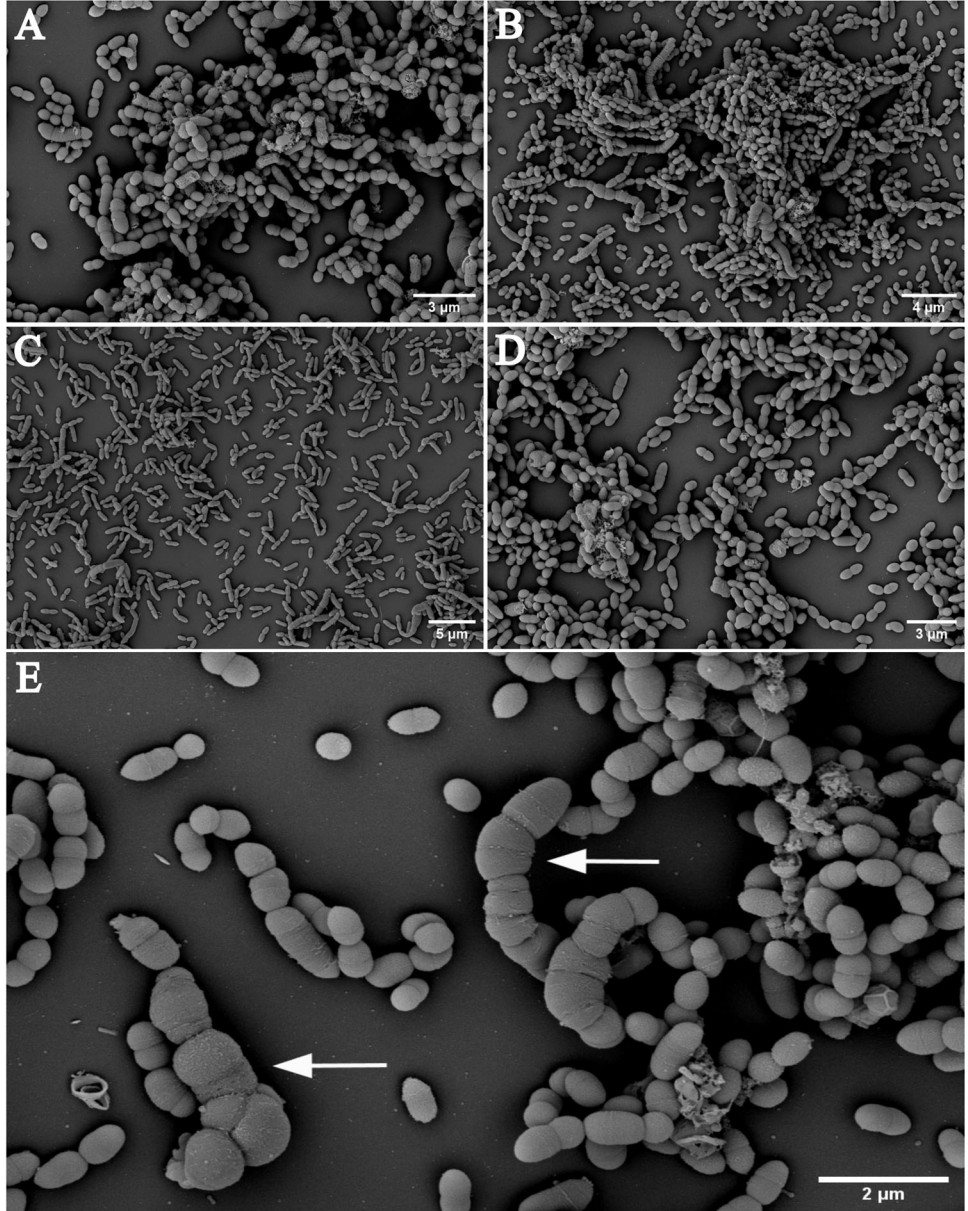

**Fig. 3 | Scanning electron micrographs visualizing the cell morphology of enriched archaeal cultures and co-enriched background bacteria. A** enrichment culture P86, predominated by *M. smithii*. **B** enrichment culture P89, predominated by *Cand*. M. intestini. **C** enrichment culture P31, predominated by *M. smithii*. **D** enrichment culture P51, presence of *Cand*. M. intestini. **E** enrichment culture P86 (MS + B.t.), predominated by *Cand*. M. intestini, enlarged for better visibility of different cell phenotypes. Some rods appear bloated (arrows).

Actually, in all found significant associations (Supplementary Fig. S8), the *Methanobrevibacter* species showed opposed preference for bacterial partners, indicating a somewhat different niche preference in the gastrointestinal setting. Also, *M. smithii* showed numerous negative correlations, whereas *Cand*. M. intestini, showed mostly positive correlations to co-cultivated bacteria (Supplementary Fig. S8).

Notably, *B. thetaiotaomicron*, although provided in the culture flasks to allow enrichment, was no longer predominant at later stages of the cultivation, indicating that the potential beneficial effect was rather on pre-priming the medium for methanogenic growth.

Although both methanoarchaeal species (*M. smithii* and *Cand*. M. intestini) co-exist in original gastrointestinal samples[3], their growth in the enrichment cultures was significantly negatively correlated (Spearman correlation, $r_s = -0.6559$, $q = 1.967059*10^{-7}$), indicating a competitive behavior, at least under the cultivation conditions applied.

## Cultured archaea represent nine novel, thus far uncultivated *Methanobrevibacter* strains

From all enrichment cultures, the respective archaeal genomes could be recovered by Nanopore sequencing and genome assembly (see "Material and Methods"). A summary of the genome classification according to the GTDB database, as well as the checkM2 output representing genome completeness and contamination, can be found in Supplementary Data S5. The obtained genomes were placed in the context of available genomes from human-associated *Methanobrevibacter* strains for further analysis[2] (list of genomes provided in Supplementary Data S6; all genome sequences are available through our GitHub repository[26]).

In general, we preserved genomes from nine *M. smithii* and eight *Cand*. M. intestini strains. From the *Cand*. M. intestini clade, we obtained overall four different strain-clusters (cutoff: 99% similarity; Fig. 4; MIC-1 to MIC-4), while five clusters were obtained from the *M.*

*smithii* clade (MS-1 to MS-5). All of these clades, except one (MIC-1) represent so far unknown genomes (below 99% identity compared to 1,167 genomes available; ANI values and matrix are given in Supplementary Data S7-S8[2];). All strains had no cultivated representative thus far, indicating that our enrichment strategy is well suited to expand the cultivable *Methanobrevibacter* diversity.

### Differences of *Methanobrevibacter smithii* and *Candidatus* Methanobrevibacter intestini isolates revealed through sequence- and structure-based annotation

*M. smithii* and *Cand.* M. intestini show various differences in their genomic inventory[2]. Based on the sequence-based annotation obtained through eggNOG profiling, our findings here corroborate this observation. Including only high-quality genomes (> 90% completeness and <10% contamination), we identified a significantly different functional profile (PERMANOVA, Ft = 2.2927, $R^2$ = 0.12533, Pr(> F) = 0.002) among our enriched *Methanobrevibacter* representatives, clearly separating the two species (Supplementary Fig. S9). The analysis included reference genomes of *M. smithii* (Gut_genome132205; DSM2374) and *Cand.* M. intestini (Gut_genome143185; WWM1085; Supplementary Data S6-7). The most discriminative genes of all obtained genomes (present in ≥ 90% or ≤ 10% of the genomes) are shown in Table 2.

As typical for archaeal genomes, a high number of proteins cannot be assigned to a function (54% across 1.8 million proteins from 1167 genomes[2]), due to missing databases and mechanistic information. For that reason, we performed a structure-based protein function prediction via DeepFRI which is reliant on Alpha-fold[30] for the most discriminative functions. By doing so, three previously unknown functional genes (2AVB5, 2C5WB and 2DDW5) from *Cand* M. intestini could be annotated (Table 2). Frequencies of the occurring GO terms in annotations are shown as word clouds in Supplementary Fig. S10.

Most discriminative functions were involved in substrate/ ion transport, such as for molybdate, as all *M. smithii* genomes analyzed in this study carried the *modA* and *modB* subunits for the ABC (molybdate) transporter. *ModA* binds molybdate and tungstate with high affinity, and together with *ModB* transport permease protein is widely distributed across methanogens. Molybdopterin is an important cofactor for the formate dehydrogenase, indicating some difference in the formate metabolism of both species[31]. Indeed, differences in the behavior of formate metabolism in *Cand.* M. intestini was found, as it, in contrast to *M. smithii* strains, accumulated formate in its medium, although the formate dehydrogenase and formate transporters are coded in its genome[32]. Further, unique nucleotide-sugar transmembrane transporters for each species were identified through DeepFri annotation.

Another finding was the potential presence of an oxalyl-CoA decarboxylase in the *Cand.* M. intestini; these enzymes are highly important to eliminate oxalates from human diet contained in e.g., coffee, tea and chocolate[33], and would allow for the release of $CO_2$ and formyl-CoA (https://www.ebi.ac.uk/QuickGO/).

Although *Cand.* M. intestini genomes were somewhat larger (1.8 Mbp) compared to *M. smithii* (1.7 Mbp; see also[2]), the overall gene count was similar (~1900 genes per genome). The percentage of core genes was also very similar (66.39% for *M. smithii*; 66.61% for *Cand.* M. intestini), but variations were observed in the percentage of non-core functions (27.79% and 29.47, respectively) (Supplementary Fig. S11). Notably, the representative *Cand.* M. intestini genome (Gut_genome132205) revealed double the amount (n = 36) of genes involved in defense mechanisms than the representative genome of *M. smithii* (Gut_genome132205, DSM2374, n = 18). The additional genes included mainly CRISPR-associated proteins or genes involved in Type I restriction modification (Supplementary Data S9), indicating a potentially more competitive environment for *Cand.* M. intestini (Supplementary Fig. S12)

### Functional annotations indicate potential differences of *Methanobrevibacter* species according donor health status

We explored the functional profiles of the archaeal cultures obtained from both healthy and diseased participants. This analysis excluded all reference genomes due to the unknown health status of the source individuals and thus only initial insights can be retrieved. Please note further, that the following predictions for genome functions were only inferred from genomic data and not tested experimentally.

Overall, *Methanobrevibacter* genomes of healthy and diseased individuals did not differ significantly (Supplementary Fig. S13, egg-NOG annotation), but the *Cand.* M. intestini population revealed variances according to the health status of the respective donor. Some potential unique functions for representatives stemming from the diseased (n = 4) and healthy cohort (n = 4) could also be identified (Supplementary Data S10). Unique functions found in *Cand.* M. intestini genomes from subjects with gastrointestinal issues were mainly CRISPR-associated proteins (arCOG05124: CRISPR-associated protein TM1802 (cas_TM1802)); COG3649: (CRISPR-associated protein Cas7)). Three unresolved annotations (arCOG07602, 2DP7E, 2DQUR) were further tackled through structure-based function prediction and identified functions involved, amongst others, hydrolase activities (Supplementary Fig. S14–15; Supplementary Data S10).

In general, the identified functions (DeepFri) for the unique COGs of *Cand.* M. intestini genomes derived from the diseased participants were mainly related to 5'–3' nuclease activity, RNA polymerase activity and DNA polymerase complex, suggesting the alteration in gene expression, with the metabolic functions being mainly organonitrogen compound biosynthesis and macromolecule metabolism (HQ annotations only; Supplementary Fig. S16).

However, the *Cand.* M. intestini genomes obtained from the healthy cohort revealed a higher number of unique functions, in particular when DeepFRI-based annotation was performed. Annotations exclusive to healthy cohorts included a wider range of functions, e.g., peptidase regulation, lipid metabolism, glycosyltransferase, methyltransferase and nucleotidyltransferase, as well as regulation of signal transduction and cell communication. Other annotations were associated with functions associated with membrane, cytoskeleton and chromatin (Supplementary Fig. S16).

In summary, genes unique to the *Cand.* M. intestini genomes preliminarily identified in the healthy cohort in this study, encode a broad range of functions (cellular processing, metabolism, information storage and processing), whereas genes unique to the diseased cohort were mainly involved in CRISPR activities, DNA and RNA processing (Supplementary Fig. S16), warranting a closer look into this matter with additional cultures and genomes.

## Discussion

Despite acknowledging archaea in the human gut microbiome for over 40 years[14], our understanding of their functions and roles are still limited. The major bottleneck is the rare availability of cultivable representatives that would allow for deeper, mechanistic, comprehensive analyzes, including multi-omics and interaction studies[34].

### Development of an efficient cultivation procedure

In this study, we developed a targeted culturing procedure to efficiently enrich and isolate methanogenic archaea from human fecal samples. This approach enabled the stable cultivation of several *M. smithii* and *Cand.* M. intestini strains from participants with varying health conditions, facilitating genomic and functional comparisons. These cultures include nine previously uncultivated strains, with only one revealing overlap with one of more than 1000 recently recovered metagenome assembled genomes[2].

From all donors, except Crohn's disease patients, stable methanogen cultures could be recovered, which is a proof of the efficient

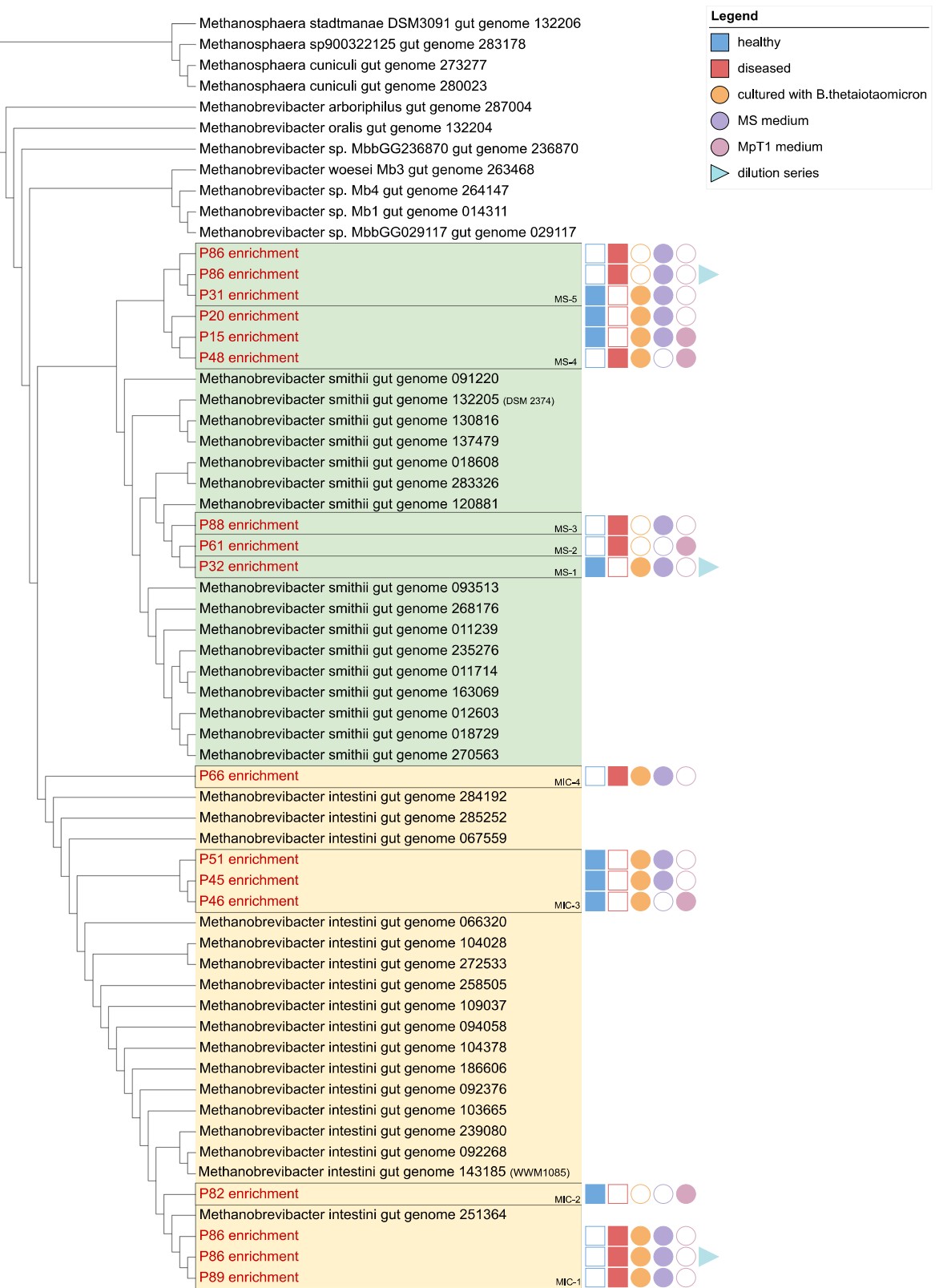

**Fig. 4 | Genome-based tree of all cultured *Methanobrevibacter*.** Hierarchical clustering based on the ANI matrix, retrieved by the comparison of all cultured *Methanobrevibacter* (red letters) and already available genomes from isolates and MAGs (Supplementary Data S6). The *M. smithii* clade is indicated with a green background color; the *Cand*. M. intestini is shown with yellow background color. Cultures are accompanied by additional information on their source (healthy/diseased donor, and culture conditions). Clusters representing one strain are indicated by black boxes (Supplementary Data S8). Source data are provided as a Source Data file.

**Table 2 | Summary of the most discriminative genes found in genomes of *M. smithii* and *Cand.* M. intestini**

| eggNOG orthologous groups | Prevalence in *M. smithii* genomes | | Prevalence in *Cand.* M. intestini genomes | | COG category | Genomic annotation (EggNOG) | Functional annotation (DeepFri) |
|---|---|---|---|---|---|---|---|
| | count (total/present) | present in % of genomes | count (total/present) | present in % of genomes | | | |
| **COG0725** | 12/12 | 100 | 0/10 | 0 | P | ABC-type molybdate transport system, periplasmic component (*modA*) | ion binding (GO:0043167)** |
| **COG4149** | 12/12 | 100 | 0/10 | 0 | P | ABC-type molybdate transport system, permease component (*modB*) | regulation of metabolic process (GO:0080090)*/**; catalytic activity (GO:0003824)*/**; membrane-associated (GO:0016020)** |
| 2EI95 | 12/12 | 100 | 1/10 | 10 | F/G | unknown function | nucleotide-sugar transmembrane transporter activity (GO:0005338) ** |
| COG4925 | 11/12 | 91.67 | 1/10 | 10 | I | sulfurtransferase activity; cyclophilin-like domain-containing protein | transferase activity (GO:0016740) |
| **COG0586** | 0/12 | 0 | 10/10 | 100 | S | Uncharacterized membrane protein DedA, SNARE-associated domain | nd |
| **COG2076** | 0/12 | 0 | 10/10 | 100 | P | Membrane transporters of cations and cationic drugs | nd |
| **arCOG00818** | 1/12 | 8.33 | 10/10 | 100 | S | AbrB family transcriptional regulator | nd |
| 2AVB5 | 0/12 | 0 | 9/10 | 90 | F/G | unknown function | nucleotide-sugar transmembrane transporter activity (GO:0005338) |
| 2C5WB | 0/12 | 0 | 9/10 | 90 | CHQ | unknown function | oxidoreductase (GO:0016491); oxalyl-CoA decarboxylase activity (GO:0008949) |
| 2DDW5 | 0/12 | 0 | 9/10 | 90 | G | unknown function | glycosyltransferase activity (GO:0016757); oxidoreductase (GO:0016616) involved in the steroid biosynthetic process (GO:0006694) |
| COG5015 | 0/12 | 0 | 9/10 | 90 | S | Pyridoxamine 5'-phosphate oxidase family | nd |
| COG1733 | 0/12 | 0 | 9/10 | 90 | K | HxlR-like helix-turn-helix | nd |

Table shows counts and percentages of genomes in which the respective function is present. Distinguishing characteristics derived from high-quality genomes are emphasized in bold.[1]
P: Inorganic ion transport and metabolism, I: Lipid metabolism, S: Function unknown, K: Transcription, F: Nucleotide transport and metabolism, G: Carbohydrate transport and metabolism, C: Energy production and conversion, H: Coenzyme transport and metabolism, Q: Secondary metabolites biosynthesis, transport and catabolism. N.d.: not done. **: High quality annotation (annotations with score ≥ 0.5); *: medium quality annotation (annotations with score ≥ 0.25).

enrichment pipeline. Such high efficiency could even allow personalized archaeome studies based on individual microbiomes.

### Growth inhibition by *C. minuta*

(Human) methanogens often rely on syntrophy with bacterial partners that provide small carbon compounds and hydrogen for methanogenesis[35]. In particular *Christensenella* and *Bacteroides* species were considered potent partners, as they generally correlate positively with *Methanobrevibacter* species in natural gut microbiomes, in previous experiments[23] and confirmation through in silico modeling[22] (and this study). In our targeted approach, we utilized this positive interaction and added both bacterial species to the culture media in order to induce the low levels of methanogens to grow, as it has been proposed earlier[34]. Nevertheless, in contrast to cultures in which *B. thetaiotaomicron* was provided as syntrophic partner, all our enrichment attempts remained methane- and methanogen-negative when *C. minuta* was added, indicating a potential competition under the artificial culture-media conditions, despite their re-current co-existence in natural gastrointestinal microbiomes[3,21]. Pre-priming of the media with *B. thetaiotaomicron*, however, was slightly advantageous, as one *Methanobrevibacter* strain could only be cultured with its addition.

### Co-enriched bacteria from human GIT microbiome

Besides those known interaction partners, we co-enriched background bacteria from feces, which might give indications on potential, natural bacterial interaction partners within the GIT microbiome. In contrast to *M. smithii* enrichments, *Cand*. M. intestini was revealing a strong co-occurrence with bacteria, such as *E. coli* and *Bacteroides* species. Both bacterial taxa produce a variety of substances from more or less complex sugar components, such as succinate, formate, acetate, lactate and ethanol[36,37]. This indicates a potential dependency of *Cand*. M. intestini on these or other components that are not well provided in the medium, or an inhibition of *M. smithii* by those or other components.

### Retrieval of pure cultures

The provision of syntrophic partners, as well as the co-enrichment of bacteria from the original samples results in a minor, but stable background contamination in the enrichment cultures. To obtain final pure cultures, this background contamination has to be removed, by e.g., using optical tweezers, micromanipulators, encapsulation methods or high-throughput single cell isolation[34]. In our case, we used a combination of antibiotics/lysozyme treatment and dilution series, which substantially improved the archaeal predominance, and meanwhile led to several pure enrichments without bacterial contamination.

As all enrichment cultures are very stable and maintained for months and > 50 transfers in our laboratory now, we also might have to consider that some co-cultures that resist clonal purification, are reliant on the archaeal-bacterial interaction, i.e., by syntrophy or any other mechanisms. Co-cultures reflect natural conditions to a greater extent, and creative solutions are needed to still obtain pure cultures[38,39], which are, for example, a prerequisite for the inclusion of these organisms in a culture collection. However, providing these strains to the scientific community is a declared goal, in order to extend the wealth of available *Methanobrevibacter* isolated from the human gut for future mechanistic studies.

### Insights into the genomes and advantages of structure-based predictions

*M. smithii* and *Cand*. M. intestini stand out as distinct species, characterized by unique microbial networks, physiological roles, and genomic potentials[28,32]. Like many other archaea, much of their genetic landscape remains unexplored, as a significant portion of their genes lack annotation, and genetic models are yet to be established. To delve deeper into their characteristics, we employed structure-based predictions to complement sequence-based tools for annotation[40]. As previously observed[41], also archaeal COG groups when further annotated using DeepFRI[30], exhibit enhanced accuracy and coverage of annotated functions compared to orthodox orthology-based methods. As such, we were able to retrieve a higher resolution in the distinction of *M. smithii* and *Cand*. M. intestini genomes, and identified some novel functions (such as the putative oxalyl-CoA decarboxylase) that could be relevant for the gastrointestinal ecosystem.

The role of methanogenic archaea for health and disease are still a matter of ongoing studies (summarized in ref. 3, and ref. 18). In particular, the role of *Cand*. M. intestini remains to be explored, as earlier studies could not resolve both species separately, due to the high sequence similarity in the 16S rRNA gene[32]. In our study, the number of participants and the subsequent cultures and genomes was quite low, so the relevance of either species for disease/health cannot be concluded at this stage. Nevertheless, structure-based analyzes of the clusters of orthologous groups of the *Cand*. M. intestini genomes revealed some differences based on the health status of the donor, with a broader variety of functions associated with the healthy phenotype, and a number of functions associated with CRISPR, and DNA/RNA processing in the diseased phenotypes, also lacking a clear indication of their subcellular location, which might indicate a potential foreign origin (e.g., phage/virus). These results should be considered as preliminary evidence, showcasing the potential lying in the cultures and genomes, but also emphasizing the need for further investigation.

### Conclusion and outlook

In this study, we have successfully extended the cultivable human archaeome through a targeted cultivation approach that relied on a strategic selection of samples and optimized culture conditions, including syntrophic bacterial strains. This approach facilitated the growth of *Methanobrevibacter* species, resulting in the stable cultivation of nine novel strains belonging to the two major archaeal species in the human gastrointestinal ecosystem, namely *M. smithii* and *Cand*. M. intestini. The derived genomes can be used to obtain first insights into the physiology and functions that might be relevant for human health or disease.

Furthermore, our method has proven highly efficient, as evidenced by ongoing work in our laboratory, as we were able to enrich methanogens even from individuals with methane content below 5 ppm in their breath. This highlights the robustness of our approach and its potential for widespread application. However, it should be noted that the success rate of obtaining enrichments was lower in low methane emitters (13% methanogen-positive enrichments) compared to a success rate of 88% in pre-screened donors.

Our work opens the way for comprehensive analyzes that promise to deepen our understanding of methanogen dynamics within the human gastrointestinal tract. Expanding the public isolate collection is essential for gaining further insights into the roles of methanogens in the human microbiome.

## Methods

### Ethics and participant recruitment

Sixteen participants with different health conditions (healthy or with any kind of gastrointestinal disorder) were recruited based on their baseline breath $CH_4$ levels (> 5 ppm / high $CH_4$ emitters). The recruitment was supported by medical doctors at the State Hospital Graz (Austria) and an independent physician also located in Graz, Austria. Participants were asked to undergo a breath test, provide a stool sample, and complete a questionnaire with health status information (Supplementary Data S1). Further, all participants signed an informed consent before participation. A list of exclusion and inclusion criteria can be found in Supplementary Data S2. The study was approved by the local ethics committee at the Medical University of Graz, Austria

(EK-Nr: 31-452 ex 18/19). The sex of participants was recorded (Supplementary Data S1) but was not included in any analysis due to the low sample number processed and a different research focus.

## Sample collection, processing and storage
For the pre-selection of participants, a breath test was performed[21]. For this purpose, participants performed the breath test either directly on a GastroCH$_4$ECK™ Gastrolyzer (Bedfont Scientific Ltd., UK) or with a GastroCH$_4$ECK™ breath bag (Bedfont Scientific Ltd., UK), which was used for a later measurement on the Gastrolyzer device. Measurements were performed according to the manufacturer's instructions. Participants, who exhaled more than 5 ppm CH$_4$, were considered as high CH$_4$ emitters based on[21] and were asked to provide a stool sample and personal data including their health status. For anaerobic cultivation the stool samples were collected with eSwabs™ (Copan, USA) and from the same source for metagenomics with a sterile stool sample container (Sarstedt, AUT). The stool sample container was pre-filled with 5 ml DNA/RNA free ethanol to stabilize the samples[42].

The participants were asked to return the samples within 24 h after collection and store them at 4 °C in the meantime. After delivery, the samples for anaerobic cultivation were immediately transferred into sterile anoxic (N$_2$) serum bottles (Glasgerätebau Ochs Laborfachhandel e.K, GER) with sterile syringes (Injekt®-F Solo, B.Braun SE, GER) and the same volume of glycerol containing ascorbic acid (50% glycerol (v/v), 50% 1x PBS (v/v), and 1 g/l ascorbic acid) was added. Afterwards, the samples were mixed by inverting, incubated for one hour at 4 °C and then stored at −80 °C until cultivation. Samples for DNA extraction were transferred in DNA-free 1.5 ml sterile reaction tubes (Eppendorf, GER) and frozen at −80 °C until further processing.

## Microbiome profiling of the original samples
The DNA of the original stool samples was isolated using the PureLink™ Microbiome DNA Purification Kit according to manufacturer's instructions (Invitrogen, Thermo-Fisher, USA).

Shotgun sequencing was conducted by the company Macrogen using the Illumina HiSeq device. Raw sequencing reads were quality filtered with ATLAS (version 2.16.2) which also included the removal of human reads[43]. For taxonomic annotation the following tools were used: Kraken2 (version 2.1.2; with a conf. 0.1) and Bracken (version 2.7; read length 100)[44,45]. To complete the hierarchical lineage information the script "kreport2mpa.py" was used that is publicly available at GitHub (https://github.com/jenniferlu717/KrakenTools/blob/master/kreport2mpa.py) before merging the feature tables of all samples.

## General workflow of enrichment and cultivation
### Selection of bacterial syntrophy partners and metabolic modeling.
Metabolic modeling was used to test the suitability of bacterial syntrophy partners on MS[46] and MpT1[47,48] media for enrichment of *Methanobrevibacter smithii* (full recipes are available in ref. 32).

Draft metabolic models were generated using gapseq (v1.2)[49] on the genomes of the *B. thetaiotamicron*, *C. minuta* and *M. smithii* (GenBank IDs NC_004663.1, GCF_001652705.1 and ABYW00000000.1, respectively). The recipes for MS and MpT1 media were used to build a representation of media constituents in the ModelSEED[50] namespace. The compositions of the undefined media ingredients yeast extract and casein hydrolysate were approximated using the definitions of yeast extract and tryptone by Marinos et al.[51]. The initial models were gap-filled with gapseq on each of the media separately, including the respective nutrient additives for *C. minuta* and *B. thetaiotamicron*.

For the simulations of the co-cultures, PyCoMo[52] was used to build community metabolic models for each pair of organisms in each medium. The respective medium with bacterial additives were applied to the models. Growth of the communities was calculated at equal abundance of bacteria and *M. smithii*. Exchange metabolites and cross-feeding interactions were calculated from solutions of flux balance analysis with the same parameters.

Cross-feeding interaction profiles (metabolites produced by organism A and consumed by organism B in a given co-culture) were compared by calculating the fraction of the intersection of cross-fed metabolites divided by the union of cross-fed metabolites in two given profiles.

### Cultivation.
A general workflow of the enrichment process is shown in Fig. 1 and detailed flowcharts for each sample can be found on our GitHub repository (https://github.com/SDMUG/Methanobrevibacter_Enrichment)[26]. For the enrichment of methanogenic archaea, stool samples were gently thawed on ice and cultivated using the following two approaches: i) First, the samples were inoculated in MS and MpT1 media supplemented with antibiotics (0.1 mg/ml penicillin G potassium salt (Carl Roth GmbH + Co. KG, GER) and 0.1 mg/ml streptomycin sulfate (Carl Roth GmbH + Co. KG, GER) to reduce bacterial growth. ii) For the second approach, 0.2 ml of stool suspension was directly inoculated into MS and MpT1 media containing bacterial cultures that had already been grown for one day. For this co-cultivation, either *Bacteroides thetaiotaomicron* (DSM 2079) or *Christensenella minuta* (DSM 22607) (purchased from Leibniz Institute DSMZ - German Collection of Microorganisms and Cell Cultures GmBH, GER) were used, as these bacteria have been described to support the growth of the methanogenic archaea[22,23], and we could also predict their supportive capacities through metabolic modeling (see below).

Inoculation for this approach followed the procedure described in ref. 23. For the cultivation, both media were prepared under anaerobic conditions. In case of cultures with *B. thetaiotaomicron*, the following supplements were additionally added before inoculation to MS and MpT1 media, following the instructions of the DSMZ for the growth of *B. thetaiotaomicron* (medium 78 + haemine + Vit K1, https://www.dsmz.de/collection/catalog/details/culture/DSM-2079): 0.2 ml hemin solution (50 mg hemin (Sigma-Aldrich, USA) in 1 ml 1 M NaOH (Carl Roth GmbH + Co. KG, GER), diluted to 100 ml with ddH$_2$O); 0.2 ml vitamin K1 solution (0.1 ml vitamin K1 (Alfa Aesar GmbH & Co KG, GER) in 19 ml 95% (v/v) ethanol (Avantor Performance Material Deventer, NLD)) and 1 ml beef extract solution [100 µg/ml] (MP Biomedicals, USA). In case of *C. minuta*, the media were supplemented with 0.4 ml D-glucose solution [500 µg/ml] (Amresco Inc., USA). All supplements were also prepared under anaerobic conditions and pressurized with 1.0 bar N$_2$.

The cultures were subjected to incubation at a temperature of 37 °C using an incubation shaker set at a shaking speed of 80 rpm. The growth of archaea was influenced by the medium and the presence of supporting bacteria. The incubation period ranged from 7 to 14 days.

The cultures from both approaches were screened for methane and F$_{420}$-positive cells. In the case of approach two, screening occurred prior to FACS sorting, as the sorting process was based on F$_{420}$ autofluorescence of the methanogens.

## CH$_4$ measurement and fluorescence microscopy
To detect the presence of methanogens, the CH$_4$ sensor BCP-CH$_4$ (BlueSens gas sensor GmbH, GER) was used to measure CH$_4$ in the gas phase of the culture bottle according to the manufacturer's manual. Any cultures with CH$_4$ levels above 2% were classified as CH$_4$ positive.

The presence of methanogens was further checked via the typical autofluorescence of coenzyme F$_{420}$. For this purpose, fluorescence microscopy (Zeiss microscope Axio Imager A1, Carl Zeiss AG, GER) with fluorescence upgrade was used with Zeiss filter set 05: BP 395-440 (excitation), FT 460 (beam splitter), LP 470 (emission) and 100x magnification (objective Plan-NEOFLUAR).

## Fluorescence activated cell sorting (FACS)
To physically enrich the methanogens within the samples, the fluorescence activated cell sorting (FACS) was used on CH$_4$-positive

cultures grown with *B. thetaiotaomicron* or *C. minuta*. Sorting was conducted via the Core Facility Flow Cytometry at the Center of Medical Research in Graz, Austria with FACSAria™ IIu system (BD Biosciences, USA)[53]. Prior to sorting, samples were prepared in an anaerobic chamber (Whitley A95 Anaerobic Workstation, Don Whitley Scientific Limited, UK) as follows: 1 ml of culture was centrifuged at 4000 × $g$ for 5 min in a 1.5 ml reaction tube (Eppendorf AG, Hamburg, GER), and the resulting cell pellet was washed twice with 500 µl 1x PBS containing 1 g/L L(+)-ascorbic acid (AppliChem GmbH, GER). The cell pellet was then resuspended in 1 ml of 1x PBS + ascorbic acid and further diluted with an additional 1 ml of 1x PBS + ascorbic acid and transferred to a Falcon® 5 ml Polystyrene Round-Bottom Tube with 35 µm Cell-Strainer Cap (Corning Science Mexico S.A. de C.V., MEX) before sorting to ensure a single cell suspension.

Cells were sorted based on the autofluorescence of coenzyme $F_{420}$ (excitation wavelength at 420 nm) based on[27]. $F_{420}$-negative and $F_{420}$-positive cells were distinguished using a 405 nm laser for excitation and a 450/50 band pass filter for detecting fluorescence emission. The presence of autofluorescent cells was observed in $CH_4$-positive cultures in contrast to its absence in $CH_4$-negative cultures. In addition, original stool samples, enrichment cultures, and pure cultures of various methanogenic archaea (*Methanobrevibacter smithii* (DSM 2375), *Methanosphaera stadtmanae* (DSM 3091), and *Methanomassiliicoccus luminyensis* (DSM 25720)) were employed for identifying the sorting gate (Supplementary Figs. S1–S4). The sort time was chosen for each culture individually and ranged between 3 and 7 min, depending on the archaeal cell count.

Sorted cells (min. 800 events, details for each sample can be found in the flowcharts on our GitHub repository[26]) were transferred into a 1.5 ml microcentrifuge tube containing 250 µl of 1x PBS + ascorbic acid and inoculated into fresh medium. Afterwards the cultures were incubated at 37 °C with 80 rpm shaking speed under anaerobic conditions.

### Antibiotics, Lysozyme, Dilution series

To reduce bacterial growth and enrich methanogens within the cultures without the addition of *B. thetaiotaomicron* or *C. minuta* and cultures after FACS, antibiotics (penicillin G potassium salt (0.1 mg/ml) and streptomycin sulfate (0.1 mg/ml)) were added to the media directly before sample inoculation. For further reduction of antibiotic-resistant bacteria, lysozyme (0.5 mg/ml; all substances: Carl Roth GmbH + Co. KG, GER) was added after each culture transfer.

For further purification of the methanogens, dilution series from $10^{-1}$ to $10^{-13}$ were performed under anaerobic conditions. Growth was usually observed up to a dilution of $10^{-7}$ and $10^{-8}$. After one week of incubation (37 °C, 80 rpm shaking speed) the $CH_4$ levels of positive growth cultures were measured and cultures were evaluated further via fluorescence microscopy.

### Sequencing and sequence data analysis

Two sequencing approaches were followed to assess the composition of the cultures. Sanger sequencing was used to identify the (almost) pure archaeal and bacterial components of enrichment cultures, whereas Nanopore whole genome sequencing was used for a detailed assessment of metagenomic composition and genome reconstructions.

**Sanger sequencing.** For Sanger sequencing, the DNA was extracted from $CH_4$-positive cultures using the MagNA Pure Compact system and MagNA Pure Compact Nucleic Acid Isolation Kit (Roche Diagnostics GmbH, GER). To prepare the samples, 1 ml of culture was centrifuged at 4000 × $g$ for 10 min, then 800 µl of supernatant was discarded and 180 µl of MagNA Pure 96 External Lysis Buffer (Roche Diagnostics GmbH, GER) was added. The extraction was performed according to the manufacturer's instructions.

For archaea, following primers were used: mcrA_forward (5′-GGH GGN GTH GGD TTY CAN CAR TA-3′) and mcrA_reverse (5′-CRT TCA TNG CRT ART TNG GRW AGT-3′) (Eurofins Genomics, Ebersberg (DE)), to amplify the *mcrA* gene of the methanogens (primers were modified from[54] and adapted to the human archaea genome catalog[2]).

The amplification was carried out using Ex Taq® DNA polymerase, 10x ExTaq Buffer w/MgCl2, 2.5 mM dNTP Mix (Takara Bio Inc., JPN), 10 µM of each primer, 20-30 ng template, and PCR- grade water (Jena Bioscience GmbH, GER). The PCR conditions were as follows: initial denaturation for 5 min at 95 °C, 35 cycles of denaturation for 40 s at 95 °C, annealing for 2 min at 51.5 °C, extension for 1 min at 72 °C, and a final extension for 10 min at 72 °C.

To identify remaining bacteria in the $CH_4$-positive cultures, the 16S rRNA gene was amplified using forward primer 9bf (5′-GRG TTT GAT CCT GGC TCA G-3′) and reverse primer 1406uR (5′- ACG GGC GGT GTG TRC AA-3′) (Eurofins Genomics, GER)[55,56]. The conditions for 16S rRNA gene PCR were as follows: initial denaturation for 2 min at 95 °C, 10 cycles of denaturation for 30 s at 96 °C, annealing for 30 s at 60 °C, extension for 1 min at 72 °C, 25 cycles of denaturation for 25 s at 94 °C, annealing for 30 s at 60 °C, extension for 1 min at 72 °C, and a final extension for 10 min at 72 °C. PCR success was determined by agarose gel electrophoresis.

Afterward, the PCR products were purified using the Monarch PCR & DNA Cleanup Kit (New England Biolabs GmbH, GER), according to the manufacturer's manual and the DNA concentration of the purified samples was measured using a Nanodrop 2000c spectrophotometer (Thermo Fisher Scientific Inc., USA). To the purified products the respective primers with a concentration of 10 µM were added and the mixture was sent to Eurofins Genomics AT GmbH, Vienna (Austria) for sequencing.

The sequencing results were analyzed and identified using the standard nucleotide BLAST from NCBI. However, the *mcrA* gene sequence of *Cand*. Methanobrevibacter intestini was not yet available in NCBI, so the taxonomic affiliation was determined by constructing a phylogenetic tree of the *mcrA* gene sequences using an in-house database (see below).

**Nanopore sequencing.** To determine the complete genome of the enriched archaea and the remaining co-existing bacteria from the examined cultures, Nanopore sequencing on MinION Mk1C (Oxford Nanopore Technologies plc., UK) was performed (nanoporetech.com). Initially, DNA was extracted (Invitrogen™ PureLink™ Microbiome DNA Purification Kit, Thermo Fisher Scientific Inc, USA), following the manufacturer's protocol. The quality and concentration of extracted DNA were assessed using a Nanodrop 2000c spectrophotometer (Thermo Fisher Scientific Inc., USA) and an Invitrogen™ Qubit™ 3 Fluorometer (Thermo Fisher Scientific Inc., USA), and DNA fragmentation was evaluated using agarose gel electrophoresis. The DNA was stored at −20 °C until further processing.

For library preparation, DNA was repaired using the NEBNext Companion Module (New England Biolabs GmbH, GER), and then prepared for the sequencing on a chemistry version 14 flow cell (R10.4.1, FLO-MIN114) following the Ligation sequencing gDNA–Native Barcoding Kit 24 V14 (SQK-NBD114.24) according to the manufacturer's instructions (nanoporetech.com).

### Sequence data analysis

*McrA* genes were aligned and integrated into an in-house database using ARB[57] and MEGA11[58]. The in-house database is based on a collection of *mcrA* gene sequences from available genomes from human gut archaea[2]. Information on the *mcrA* gene of $CH_4$-bearing enrichments was mainly used to identify unique enrichments and to weed-out isolate-duplicates at early stages. For that, distance matrices were calculated, integrating the generated sequences into the database. It should be mentioned that *M. smithii* and *Cand*. M. intestini cannot

clearly be distinguished based on the 16S rRNA gene sequence, but the *mcrA* gene allows for a clear distinction of both species[32].

The obtained data from Nanopore sequencing was analyzed using the following configurations: the MinION Mk1C device was set to run for 72 h, pore scan frequency of 1.5 h, minimum read length of 200 bp, high-accuracy base calling, and enabled active channel selection, reserved pores, read splitting, trimming barcodes and mid-read barcode filtering. The software versions used included MinKNOW v22.10.7, Bream v7.3.5, Configuration v5.3.8, Guppy v6.3.9, and MinKNOW Core v5.3.1 (nanoporetech.com). Obtained sequencing data from Nanopore was analyzed using the following specifications: Simplex base calling was carried out on a GPU node at the Life Science Compute Cluster (LiSC) UniversityofVienna (Austria),usingguppy-gpuv6.4.2andthe dna_r10.4.1_e8.2_260bps_sup.cfg model for superior base calling. Duplex base calling was achieved by following the recommended tutorial (https://github.com/nanoporetech/duplextools) using dorado v0.2.3 with the dna_r10.4.1_e8.2_260bps_fast@v4.1.0 fast model, followed by finding duplex pairs with duplextools v0.2.20-3.10 and dorado stereo/duplex base calling with the dna_r10.4.1_e8.2_260bps_sup@v4.1.0 superior model.

Further analysis of the merged simplex and duplex reads involved quality control using NanoPlot nanocomp v1.20.0[59], filtering with filtlong v0.2.1 (--min-length 200;--keep_percent 90, and --target_bases 20 million) (https://github.com/rrwick/Filtlong).

The filtered reads were then assembled using flye v2.9.1 in --nano_hq and --meta mode with a target genome size of 2 Mbp[60]. The reads were mapped using minimap2 v2.24[61], and then read polishing was performed with racon v1.5.0 (https://github.com/isovic/racon), using the following settings: (-m -8 -x -6 -g -8 -w -500). A consensus contig was obtained using medaka v1.7.2 and the r1041_e82_260bps_sup_g632 model (https://github.com/nanoporetech/medaka). Basic genome statistics were generated using genometools calling gt seqstat[62], and the genomes were classified using gtdbtk v2.2.6 with the --full_tree option enabled[63]. Genome completeness and contamination were estimated using checkm2 v1.0.1 with –allmodels[64].

## Taxonomic Classification
Reads were mapped against the Unified Human Gastrointestinal Genome (UHGG v.2.0.1) downloaded from MGnify (https://www.ebi.ac.uk/metagenomics) consisting of more than 289.000 archaeal and bacterial RefSeq genomes using Kraken v.2.1.2[45]. In order to increase the specificity and to compensate for the chance of returning the incorrect lowest common ancestor (LCA) of all genomes, a confidence threshold of 0.3 was chosen for Kraken2. To determine the relative abundance of bacterial and archaeal species, the Kraken2 output was subjected to analysis using Bracken v.2.7. (https://ccb.jhu.edu/software/bracken/) with default settings.

## Scanning electron microscopy (SEM)
The cell morphology of the enriched archaea and their remaining co-existing bacteria was visualized using the Zeiss Sigma 500 VP scanning electron microscope (Carl Zeiss AG, GER). Culture samples of 2 ml (with the supernatant not removed) were centrifuged at $4000 \times g$ for 10 min, and the resulting pellets were handed over to the Core Facility Ultrastructure Analysis from Medical University of Graz (Austria) for SEM preparation. Briefly, the cells were mounted on coverslips, fixed with 2% (wt/vol) paraformaldehyde in 0.1 M phosphate buffered saline, pH 7.4 and 2.5% glutaraldehyde in 0.1 M phosphate buffered saline, pH 7.4, and dehydrated stepwise in a graded ethanol series. Samples were post fixed with 1% osmium tetroxide (Electron microscopy Sciences) for 1 h at room temperature and subsequently dehydrated in graded ethanol series (30-96% and 100% (vol/vol) EtOH). Further, HMDS (Merck | Sigma - Aldrich) was applied. Coverslips were placed on stubs covered with a conductive double coated carbon tape[65]. SEM micrographs were taken with an electron voltage of 5 keV and a magnification of 3–21 kX, using the secondary electron detector (Zeiss Oberkochen).

## Tree calculation, correlations and pangenome analysis
The ANI (average nucleotide identity) matrix of the genomes (all-vs-all) was calculated using the ANI Matrix calculator provided through (http://enve-omics.ce.gatech.edu/g-matrix/); the resulting phylogenetic tree was annotated using itol[66].

Correlation heatmaps were generated in R. All scripts are provided in our GitHub repository (https://github.com/SDMUG/Methanobrevibacter_Enrichment)[26]. Statistical associations were based on spearman correlations, which are also included in the provided script.

Pangenome analyzes were performed using kbase[67] based on RastK annotation, the Compute Pangenome and Pangenome Circle Plot (v1.2.0) function, including all available high-quality (> 90% completeness and < 10% contamination) *M. smithii* and *Cand*. M. intestini genomes (this study and[2]), with genomes of Gut_genome132205 and Gut_genome143185 serving as reference genomes.

## Genomic, functional annotation and prediction using DeepFRI
Genome comparison was performed using drep v3.4.2[68], and genome annotation was carried out using eggnog-mapper v2.1.10 (default settings, e-value cut-off 0.001)[69–74]. Prodigal was used for gene identification.

To examine the functional differences between *M. smithii* and *Cand*. M. intestini as well as to explore the differences between functions related to a specific health condition, PCoA were performed using a Bray-Curtis distance matrix. The visualization was done using the R package ggplot2 (v. 3.4.4)[75]. PERMANOVA was performed with $n = 999$ permutations to identify statistical significance in the functional profiles.

For identifying the functions of protein-coding genes in archaeal genomes, protein-coding genes were first predicted using Prodigal (https://github.com/hyattpd/Prodigal). Then, gene sequences translated to amino acids were annotated using a metagenomic pipeline (https://github.com/bioinf-mcb/Metagenomic-DeepFRI/) with an embedded DeepFRI tool[30]. The pipeline generated a query contact map using results from mmseqs2 target database (https://github.com/soedinglab/MMseqs2) searching for similar protein sequences with known structures. Next, the contact map alignment was performed to use it as input to DeepFRI Graph Convolutional Network (GCN) and annotations without matches to known structures were processed with Convolutional Neural Network (CNN). DeepFRI version 1.1 was used in this study; in version 1.1 we retrained the original DeepFRI architecture using high-quality models from the AlphaFold Database, thus increasing the number of functions the method can predict by up to 4x[76]. For medium quality (MQ) and high quality (HQ) annotation, scores higher than or equal to 0.3 and 0.5 were used, respectively. For selected genes, a structure prediction was performed using AlphaFold database[76,77], followed by DeepFRI annotation (https://github.com/SDMUG/Methanobrevibacter_Enrichment). Wordclouds were generated using the pip package (https://pypi.org/project/wordcloud/).

## Data availability
All used and created data is made publicly available on our GitHub repository: https://github.com/SDMUG/Methanobrevibacter_Enrichment[26] (permanent link: DOI: 10.5281/zenodo.13153860). Metagenomic datasets are available through EMBL ENA: PRJEB75175. Source data are provided with this paper for Fig. 2, Fig. 4 and Supplementary Figs. Source data are provided in this paper.

## Code availability
All codes are provided in our GitHub repository[26]: https://doi.org/10.5281/zenodo.13153860.

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

## Acknowledgements

We want to acknowledge the support of Petra Jurše Kupper and Astrid Koller in the recruitment process, and we thank all donors who have contributed with their samples. This research was funded in whole or in part by the Austrian Science Fund (FWF) [grants P 32697, P 30796, SFB F-83, COE 7, doc. funds project DOC 69, given to CME]. For open access purposes, the author has applied a CC BY public copyright license to any author-accepted manuscript version arising from this submission. SD and MP were supported by their local doctoral programs MolMed (Medical University of Graz) and the Doctoral School in Microbiology and Environmental Science (University of Vienna).

## Author contributions

The study was designed by S.D. and C.M.E. Method establishment was done by S.D., M.B., and A.E. Recruitment was carried out by S.D. and supported by L.S., C.H., and A.M.M. Sample collection and sample preparation was performed by S.D., S.V., L.S., and T.Z. The samples were cultivated by S.D., S.V., and M.D. FACS was done by S.D., S.V., M.D. and V.Z. A.M., C.K., S.V., and M.D. performed Nanopore sequencing, and S.D., A.M., Ł.S. and C.M.E. did data analysis. S.D., S.V., M.D., V.W., and C.M.E. created the shown figures and micrographs. D.K. and D.P. prepared the samples for M.P., and T.R. performed metabolic modeling, Ł.S. and T.Ko. performed the DeepFRI analysis. S.D. and C.M.E wrote the manuscript, and Ł.S., S.V., T.Ko., T.K., R.M., R.S., M.P., and A.M. contributed to the writing of the manuscript, which was read and approved by all authors.

## Competing interests

The authors declare no competing interests.
