## [Peer Review File · Nature Communications]

Targeted isolation of Methanobrevibacter strains from fecal samples expands the cultivated human archaeomeREVIEWER COMMENTS

Reviewer #1 (Remarks to the Author):

Congratulations to this very nice study, which I enjoyed reading. The manuscript is clearly written and the conclusions are adequate.

Thank you for attempting to isolate novel methanogenic archaea from the GIT, which in one case has been successfully performed. With these novel isolates of methanogenic archaea from the GIT that you have at hand now – or of the hopefully many new isolates that you will be able to isolate in pure culture soon – many new insights into their physiology and into their putative medical implications might be possible.

Congratulations to the successful application and detailed description of the co-enrichment strategy that you have applied.

My criticism (lines 126 to 128 and 764 to 767 and elsewhere):

It is rational to enrich and isolate methanogenic archaea from the GIT of high methane exhaling individuals. However, is there any indication why individuals that exhale lower amounts of methane (let's say 1-4.99 ppm) would not be suitable to attempt enrichment and isolation? Another question in this regards is: would different methanogenic archaeal groups be responsible for lower amounts of methane in the breath? Or would it just be more difficult to enrich methanogenic archaea from stool samples from individuals with lower than 5 ppm methane in the breath? Was there any other reason to attempt enrichments of methanogenic archaea from stool samples of methane exhaling individuals with higher than 5 ppm methane in the breath?

Moreover, you mentioned that through the enrichment and isolation pipeline you have established a personalized archaeal microbiome enrichment and characterization might become possible. Which other groups of archaea did you identify among the archaeal stool community – or do you only refer to methanogenic archaea? How fast is the pipeline for enrichment of the two methanogenic groups approximately? How reproducible is your pipeline? Did you attempt to enrich the same methanogens again from the same stool sample? The pipeline for prediction of co-cultivation with bacteria was 50% successful? Any indications how to optimize the co-enrichment pipeline? The pipeline that you have established seems to be a number of reasons selective for the specific methanogens that

you could enrich. Did you attempt to isolate e.g. Methanomassiliicoccales from the GIT of the humans by using a modified version of your pipeline?

Tables and Figures

The header of Tables 1 and Table 2 are very long. Please consider reducing the header length to a one-liner and put the rest of the text as a table footnote.

References

The citations and the references are not linked. The citations in the running text are not linked in the list of references. Thus, this reviewer was not able to examine the appropriateness of the citations.

Reviewer #2 (Remarks to the Author):

The manuscript reports the successful cultivation of nine Methanobrevibacter strains from human stool samples. The presented approach includes a donor screening via CH₄ exhalation, the supply of bacterial syntrophic partners in customized media, enrichment via FACS, antibiotic treatments, serial dilution, and a close monitoring of CH₄ production. Using long-read sequencing, the author could identify encoded metabolic features and suggest potential disease associations.

Overall, this study represents an important contribution to our current understanding and future exploration of Archaea in the human gut microbiome by providing stable enrichment cultures of previously uncultivated Methanobrevibacter strains, as well as a pure isolate of a novel *M. intestini* strain.

I therefore recommend the publication of this article, and my main remarks regard the clarification of some of the results, and their presentation and interpretation.

Donor screening via CH₄ exhalation seems to be an excellent approach to preselect gut microbiomes with higher methanogen loads, but I am missing a clear presentation of results and conclusion here. Did targeting samples from donors with high CH₄ exhalation indeed yield to a better recovery of *Methanobrevibacter* strains? And if this is not the case, as suggested by the failure of some positive breath test samples to yield cultures and the fact that cultures could be obtained from donors with a methane content below 5 ppm in their breath, then can we conclude that donor screening via CH₄ exhalation is not instrumental for the presented target isolation approach?

The conclusion that the “addition of *B. thetaiotaomicron* demonstrated a beneficial impact” should also be better supported. How many positive enrichments were achieved with, versus without, the presence of *Bacteroides thetaiotaomicron*? – this is not clear to me from the results presented starting at line 480. Was the recovery of strain (P66), reported as not obtained in the abiotic medium, the only benefit from using *B. thetaiotaomicron*? The authors should present these results in more detail and include the statement from line 520 that these cocultures resulted in a “faster growth of *Methanobrevibacter*” and support the statement with the underlying data.

Figure 2 should be improved, by removing tiny, none decipherable fractions from the pie charts. It would be great to also show the relative abundances after the dilution, maybe even in the same figure, to compare the composition of the “final” (co)cultures.

Please find my detailed comments, including questions about the phylogeny, below.

Finally, during the review process, I came across the preprint titled “Expanding the cultivable human archaeome: *Methanobrevibacter intestini* sp. nov. and strain *Methanobrevibacter smithii* “GRAZ-2” from human feces” that has been submitted by the same authors. I assume this preprint is complimentary to this manuscript?

Detailed comments:

Abstract

Since enrichment is the first step in the presented approach, followed by isolation, the sentence “Our study presents a targeted cultivation method enriching methanogenic archaea from human fecal samples” should be updated to “Our study presents a method for the targeted enrichment and isolation of methanogenic archaea from human fecal samples.”

Methods

Line 186 the formatting is off

Line 207: I assume that for the 2nd approach, a certain weight/volume of the stool samples was added directly to the already growing cultures of potential syntrophic bacteria, correct? Or were the stool samples first inoculated into sterile media and then, after a certain time period, combined with the media containing syntrophs? This should be explained in a bit more detail.

Line 239 “typical autofluorescence” might be the better term

Line 260: Was the sort performed under anaerobic conditions, and if not, how did the oxygen exposure during sorting impact the prokaryotic survival rate?

Line 274: Were the cells incubated under anaerobic conditions?

Results

Line 391 “FACS was established ...” These two sentences read more like a material & methods section.

Table 1: The “Cultures” column is a bit confusing, I would rename it to “Enrichment cultures (medium/ syntroph)” or similar.

Line 404 “After initial pre-selection for unique methanoarchaea (according to their mcrA gene sequence) of all CH₄-positive cultures” – how was this sequence-based pre-selection performed, via a targeted PCR and sequencing? This workflow seems to be missing in the methods.

Fig. 2 What is the white fraction in the pie chart that takes up almost 2/3 and 1/2 in P32 and P46, respectively?

Fig. 4 Please clarify how the phylogenetic tree was inferred. The figure says it is a maximum likelihood tree, and in the methods, you mention that an ANI matrix was calculated, but there are no details provided on how the tree was inferred. Did you infer a Maximum Likelihood tree from a distance matrix? To my knowledge, ML trees require primary sequence data and cannot be inferred from a matrix. However, I am happy to learn about a new way to do so, if it exists, but please explain your tree building steps accordingly.

Line 648 You should consider the number of isolates that goes into your comparison of healthy vs diseased individuals. If, for example, more *M. intestini* isolates/genomes were obtained from healthy individuals, then it is, to be expected that “the *Cand. M. intestini* genomes obtained from the healthy cohort revealed a higher number of unique functions”, since every new strain/genome will likely contribute new/unique encoded functions.

Also, you should emphasize that these are inferred or potential functions, since the described results are based on genomics.

References

Unfortunately, the references are not numbered in my version of the manuscript, so I could not check the citations for accuracy.

Data availability

Excellent, via GitHub repository!

Reviewer Comments Enrichment

REVIEWER COMMENTS

We are grateful to the Reviewers for their valuable input and constructive feedback. We have endeavored to address their comments and concerns in this revised manuscript.

Reviewer #1 (Remarks to the Author):

Congratulations to this very nice study, which I enjoyed reading. The manuscript is clearly written and the conclusions are adequate.

Thank you for attempting to isolate novel methanogenic archaea from the GIT, which in one case has been successfully performed. With these novel isolates of methanogenic archaea from the GIT that you have at hand now – or of the hopefully many new isolates that you will be able to isolate in pure culture soon – many new insights into their physiology and into their putative medical implications might be possible.

Congratulations to the successful application and detailed description of the co-enrichment strategy that you have applied.

My criticism lines 126 to 128 and 764 to 767 and elsewhere:

- It is rational to enrich and isolate methanogenic archaea from the GIT of high methane exhaling individuals. However, is there any indication why individuals that exhale lower amounts of methane (let's say 1-4.99 ppm) would not be suitable to attempt enrichment and isolation?

Thank you for your insightful question. Yes, there are reasons why individuals with lower methane exhalation might be less suitable for the enrichment and isolation of methanogenic archaea. Individuals with lower methane exhalation have a significantly reduced abundance (1000x, see Kumpitsch et al., 2021) of methanogenic archaea in their gastrointestinal tract, making the enrichment and isolation process probably more challenging.

Despite this challenge, we have tested our approach on low methane emitters (below 5 ppm) and were able to enrich methanogenic archaea also from these individuals, although with less efficiency compared to high methane emitters. We have mentioned this also in our Conclusion and Outlook Section **Line 755** and added more details to highlight the benefit of pre-screening.

- Another question in this regards is: would different methanogenic archaeal groups be responsible for lower amounts of methane in the breath? Or would it just be more difficult to enrich methanogenic archaea from stool samples from individuals with lower than 5 ppm methane in the breath? Was there any other reason to attempt enrichments of methanogenic

archaea from stool samples of methane exhaling individuals with higher than 5 ppm methane in the breath?

Thank you for this interesting question. Indeed, our initial idea to use high-methanogen phenotypes' samples was driven by the mere increased abundance of methanogens. Ongoing studies in our lab, however, indicate indeed that different methanogenic archaeal groups are responsible for high and low methane formation, but this was not known at the time point when the enrichments were started.

Please see this biorxiv article for more information:
<https://www.biorxiv.org/content/10.1101/2024.02.09.579604v1>

- Moreover, you mentioned that through the enrichment and isolation pipeline you have established a personalized archaeal microbiome enrichment and characterization might become possible. Which other groups of archaea did you identify among the archaeal stool community – or do you only refer to methanogenic archaea? How fast is the pipeline for enrichment of the two methanogenic groups approximately?

Regarding the establishment of a personalized archaeal microbiome enrichment and characterization pipeline, our focus was primarily on methanogenic archaea, and even more focused on *Methanobrevibacter*. However, our methods are potentially adaptable, to identify and enrich other archaeal groups present in the stool community, depending on their abundance and activity. Regarding the speed, typically the enrichment process can be completed within a few weeks, from initial sample collection to obtaining enriched cultures ready for characterization, a detailed picture of the time frames can be found for each individual sample here: https://github.com/SDMUG/Methanobrevibacter_Enrichment/blob/main/Flowcharts_all%20cultures.pdf.

- How reproducible is your pipeline? Did you attempt to enrich the same methanogens again from the same stool sample?

Thank you for bringing this up. Yes, we have attempted to enrich methanogens from the same stool sample using our different approaches as shown in **Figure 1 (page 9)** and explained in **line 201ff**. In these repeated attempts, we successfully enriched the same methanogenic archaea (screened based on the *mcrA* gene; see **Supplementary file page 4**), confirming the reproducibility of our enrichment and isolation procedures.

- The pipeline for prediction of co-cultivation with bacteria was 50% successful? Any indications how to optimize the co-enrichment pipeline? The pipeline that you have established seems to be a number of reasons selective for the specific methanogens that you could enrich. Did you attempt to isolate e.g. Methanomassiliicoccales from the GIT of the humans by using a modified version of your pipeline?

By obtaining more enrichments and analyzing which bacterial co-partners persist, we can gain better insights into preferred interaction partners as discussed on **page 31, line 701ff**. This knowledge could

help optimize the enrichment process for other archaea in addition to *Methanobrevibacter*. However, our current focus has been on this specific group.

We are also intrigued by the false prediction for *C. minuta*, which has been described as a potent partner for *Methanobrevibacter*. However, under our lab conditions, all samples remained methane and methanogens-negative when co-cultured with *C. minuta*, indicating no significant support for *Methanobrevibacter* in our specific setting. Factors such as toxins or secondary metabolites, which are not included in our models, might influence this outcome. Indeed, further research is necessary to optimize the co-enrichment pipeline.

Tables and Figures

- The header of Tables 1 and Table 2 are very long. Please consider reducing the header length to a one-liner and put the rest of the text as a table footnote.

Thank you for bringing this to our attention. As recommended by the Reviewer, we have shortened the table headers and moved the additional text to the table footnotes.

References

- The citations and the references are not linked. The citations in the running text are not linked in the list of references. Thus, this reviewer was not able to examine the appropriateness of the citations.

We apologize for the oversight regarding the missing links in the references. This issue has been resolved, and we regret any inconvenience this may have caused.

Reviewer #2 (Remarks to the Author):

The manuscript reports the successful cultivation of nine *Methanobrevibacter* strains from human stool samples. The presented approach includes a donor screening via CH₄ exhalation, the supply of bacterial syntrophic partners in customized media, enrichment via FACS, antibiotic treatments, serial dilution, and a close monitoring of CH₄ production. Using long-read sequencing, the author could identify encoded metabolic features and suggest potential disease associations.

Overall, this study represents an important contribution to our current understanding and future exploration of Archaea in the human gut microbiome by providing stable enrichment cultures of previously uncultivated *Methanobrevibacter* strains, as well as a pure isolate of a novel *M. intestini* strain.

I therefore recommend the publication of this article, and my main remarks regard the clarification of some of the results, and their presentation and interpretation.

- Donor screening via CH₄ exhalation seems to be an excellent approach to preselect gut microbiomes with higher methanogen loads, but I am missing a clear presentation of results and conclusion here. Did targeting samples from donors with high CH₄ exhalation indeed yield to a better recovery of Methanobrevibacter strains? And if this is not the case, as suggested by the failure of some positive breath test samples to yield cultures and the fact that cultures could be obtained from donors with a methane content below 5 ppm in their breath, then can we conclude that donor screening via CH₄ exhalation is not instrumental for the presented target isolation approach?

Thank you for this comment. We achieved 14 positive enrichments from 16 high methane emitter samples, demonstrating the high efficiency of our cultivation approach with pre-screened donors. Individuals with lower methane exhalation have a significantly reduced abundance of methanogenic archaea in their gastrointestinal tract (1000x lower, see Kumpitsch et al., 2021), making the enrichment and isolation process more challenging. As stated in **lines 755ff**, we tested our approach with stool samples from low methane emitters. Although we were also able to enrich archaea from these samples, the efficiency was lower, with only 2 positive enrichments from 16 low methane emitter samples. We have added more details to highlight the benefit of pre-screening as well.

- The conclusion that the “addition of *B. thetaiotaomicron* demonstrated a beneficial impact” should also be better supported. How many positive enrichments were achieved with, versus without, the presence of *Bacteroides thetaiotaomicron*? – this is not clear to me from the results presented starting at line 480. Was the recovery of strain (P66), reported as not obtained in the abiotic medium, the only benefit from using *B. thetaiotaomicron*? The authors should present these results in more detail and include the statement from line 520 that these cocultures resulted in a “faster growth of *Methanobrevibacter*” and support the statement with the underlying data.

Thank you for your comment. Yes, it is true. The addition of *B. thetaiotaomicron* slightly enhanced the success of our enrichment attempts, leading to 14 positive enrichments from 16 samples, compared to 13 positive enrichments without it. While this shows only a slight improvement, this approach allowed us to obtain P66 which was not obtainable with the abiotic medium only and further did not influence the success for all other enrichments. In contrast, all enrichment attempts with *C. minuta* remained methane- and methanogen-negative, despite both bacteria being considered supportive of methanogen growth.

However, we are very thankful for your constructive feedback on this regard and have rephrased these sections in **lines 481ff.**

Regarding the growth, we are really thankful for this comment. Unfortunately, we do not have comprehensive underlying data from the cultivation attempts aside from some CH₄ measurements. The faster growth we observed, led us to select 12 of the 14 positive enrichments with *B. thetaiotaomicron* for nanopore sequencing. However, we recognize that this data is insufficient to conclusively support the statement about faster growth. Therefore, we have removed this part from and kept the focus on the indication of a potential pre-priming effect of *B. thetaiotaomicron*, as indicated in **lines 481ff.**

- Figure 2 should be improved, by removing tiny, none decipherable fractions from the pie charts. It would be great to also show the relative abundances after the dilution, maybe even in the same figure, to compare the composition of the "final" (co)cultures.

Thank you for this constructive feedback. We have improved the figure as suggested by the reviewer and adapted the figure description.

As mentioned in the manuscript in **lines 436** and **elsewhere**, the work in our lab is ongoing and in the meantime, we have improved the purity of more enrichments, we have decided to additionally provide this information also on our github repository (https://github.com/SDMUG/Methanobrevibacter_Enrichment/tree/main/Nanopore%20Sequencing).

- Please find my detailed comments, including questions about the phylogeny, below.
- Finally, during the review process, I came across the preprint titled "Expanding the cultivable human archaeome: Methanobrevibacter intestini sp. nov. and strain Methanobrevibacter smithii "GRAZ-2" from human feces" that has been submitted by the same authors. I assume this preprint is complimentary to this manuscript?

Yes, it is complementary. The referenced manuscript is about the formal description of Methanobrevibacter intestini, and does not at all overlap with the content presented herein.

Detailed comments:

Abstract

- Since enrichment is the first step in the presented approach, followed by isolation, the sentence "Our study presents a targeted cultivation method enriching methanogenic archaea from human fecal samples" should be updated to "Our study presents a method for the targeted enrichment and isolation of methanogenic archaea from human fecal samples."

Thank you for your suggestion. We agree with your recommendation and have updated the sentence (Line 42).

Methods

- Line 186 the formatting is off

Thank you for bringing this to our attention. We have corrected the formatting issue.

- Line 207: I assume that for the 2nd approach, a certain weight/volume of the stool samples was added directly to the already growing cultures of potential syntrophic bacteria, correct? Or were the stool samples first inoculated into sterile media and then, after a certain time

period, combined with the media containing syntrophs? This should be explained in a bit more detail.

Thank you so much for addressing this. Yes, it is correct we have added the stool samples directly to the already growing cultures of potential syntrophic bacteria. We rewrote this part to make it clearer, **line 208**.

- Line 239 “typical autofluorescence” might be the better term

Thank you. We have updated it (**line 242**).

- Line 260: Was the sort performed under anaerobic conditions, and if not, how did the oxygen exposure during sorting impact the prokaryotic survival rate?

Thank you for your insightful comment. The sorting was indeed performed under aerobic conditions. However, we ensured that all cultures were kept under anoxic conditions for as long as possible and were inoculated into fresh medium as quickly as possible. Pausan et al. demonstrated that methanogens can survive in oxic conditions for at least 6 hours, so the brief aerobic sorting process, which only lasted a few minutes, was not considered problematic.

Additionally, we established and tested the sorting procedure using pure archaeal cultures before applying it to enrichment cultures, and all cultures demonstrated re-growth after sorting. The details of the FACS testing are described in lines 248ff, and the associated FACS sorting images can be found in the **supplementary material on pages 6-8**.

- Line 274: Were the cells incubated under anaerobic conditions?

Yes, the cells were indeed incubated under anaerobic conditions. Thank you for pointing this out; we have added this information to the manuscript to clarify this point (**lines 276**).

Results

- Line 391 “FACS was established” These two sentences read more like a material & methods section.

Thank you for your feedback. We have revised the sentences to better fit the narrative flow of the results section (**lines 393**). We hope this revision meets your expectations.

- Table 1: The “Cultures” column is a bit confusing, I would rename it to “Enrichment cultures (medium/ syntroph)” or similar.

Thank you for your recommendation. We have made the changes accordingly .

- Line 404 “After initial pre-selection for unique methanoarchaea (according to their mcrA gene sequence) of all CH₄-positive cultures” – how was this sequence-based pre-selection performed, via a targeted PCR and sequencing? This workflow seems to be missing in the methods.

Thank you for this comment. This information can be found in our **supplementary material (see page 2-4)**. We performed targeted PCR and Sanger sequencing of the mcrA gene to screen for unique methanoarchaea. In some samples, we obtained the same culture (based on the mcrA gene) using several approaches, as shown in Figure 1. Detailed information for each sample and where we obtained replicates of the methanoarchaea, can be found in the flowcharts available on our GitHub repository:

https://github.com/SDMUG/Methanobrevibacter_Enrichment/blob/main/Flowcharts_all%20cultures.pdf.

- Fig. 2 What is the white fraction in the pie chart that takes up almost 2/3 and 1/2 in P32 and P46, respectively?

Thank you so much for bringing this up. The white fractions indicated the overall bacterial "contamination" and the composition was listed quite small next to the pie charts. Indeed, the figure was not optimal in this regard, and therefore we have adapted and simplified **Figure 2** as suggested.

- Fig. 4 Please clarify how the phylogenetic tree was inferred. The figure says it is a maximum likelihood tree, and in the methods, you mention that an ANI matrix was calculated, but there are no details provided on how the tree was inferred. Did you infer a Maximum Likelihood tree from a distance matrix? To my knowledge, ML trees require primary sequence data and cannot be inferred from a matrix. However, I am happy to learn about a new way to do so, if it exists, but please explain your tree building steps accordingly.

This is correct, thank you for pointing us to this issue. Indeed it is not a ML tree, but based on hierarchical clustering of the ANI matrix. We have corrected the information in the Figure legend accordingly. It was correct in the Materials and Methods (**Line 541**).

- Line 648 You should consider the number of isolates that goes into your comparison of healthy vs diseased individuals. If, for example, more *M. intestini* isolates/genomes were obtained from healthy individuals, then it is, to be expected that “the *Cand. M. intestini* genomes obtained from the healthy cohort revealed a higher number of unique functions”, since every new strain/genome will likely contribute new/unique encoded functions.

This is indeed an important point! For the comparisons we used only *Cand. M. intestini* strains (four from healthy, and four from diseased), to avoid biases. Nevertheless, the found differences are only indications, and as you mention in the next question, the found functions are mostly inferred and not tested yet.

- Also, you should emphasize that these are inferred or potential functions, since the described results are based on genomics.

We tried to be very careful in making clear that the functions we describe are only inferred. However, we added another sentence in **lines 621ff** to clarify once more.

References

- Unfortunately, the references are not numbered in my version of the manuscript, so I could not check the citations for accuracy.

We apologize for the oversight regarding the numbering of the references. We have resolved this issue, and the references are now correctly numbered. We regret any inconvenience this may have caused.

Data availability

Excellent, via GitHub repository!